# Shifting Compasses: A Qualitative Study of Lived Experiences Driving Perioperative Nurses to Leave the Profession Post COVID-19

**DOI:** 10.3390/healthcare13040391

**Published:** 2025-02-11

**Authors:** Amalia Sillero Sillero, Maria Gil Poisa, Sonia Ayuso Margañon, Elena Marques-Sule, Raquel Ayuso Margañon

**Affiliations:** 1Escola Universitària Gimbernat, Adscrita a la Universitat Autònoma de Barcelona (UAB), Sant Cugat del Vallès, 08174 Barcelona, Spain; amalia.sillero@eug.es; 2Social Determinants and Health Education Research Group (SDHEd), Hospital del Mar Research Institute, 08003 Barcelona, Spain; rayuso@esimar.edu.es; 3Hospital del Mar Nursing School (ESIHMar), Universitat Pompeu Fabra-Affiliated, 08003 Barcelona, Spain; 4North Florida Primary Health Care Center, Hospitalet de Llobregat, 08905 Barcelona, Spain; 5Department of Public Health, Mental Health, and Maternal and Child Health, Faculty of Nursing, University of Barcelona, 08907 Barcelona, Spain; 6Physiotherapy in Motion, Multispeciality Research Group (PTinMOTION), Faculty of Physiotherapy, Department of Physiotherapy, University of Valencia, 46010 Valencia, Spain; elena.marques@uv.es

**Keywords:** perioperative nursing, COVID-19, professional burnout, organizational culture, ethical dilemmas, qualitative research

## Abstract

**Background**: During the COVID-19 pandemic, perioperative nurses faced extraordinary demands in frontline roles, leading many to leave their positions. This study investigates the factors influencing their decisions to resign or change roles during or after the pandemic, providing insights into systemic, ethical, and emotional contributors to professional attrition. **Methods**: A qualitative study was conducted at a university hospital in Spain between December 2021 and March 2022. A hermeneutic phenomenological approach was used to analyze the lived experiences of perioperative nurses who worked during the pandemic and subsequently resigned or changed role. Data were collected through in-depth individual interviews and analyzed using Atlas.ti (version 22). Ethical issues such as informed consent and participants confidentiality were upheld. **Results**: Eighteen perioperative nurses participated. Four themes emerged: (1) balancing professional duty and personal limits, (2) the role of workplace culture (emphasizing peer support and managerial neglect), (3) resilience and moral conflict, and (4) the emotional cost of caring. **Conclusions**: Attrition among perioperative nurses during the COVID-19 pandemic was driven by physical and emotional exhaustion, lack of managerial support, ethical dilemmas, and emotional trauma. Healthcare organizations should implement strategies such as strengthening leadership, providing mental health resources, and creating a supportive work culture to improve staff retention and ensure workforce sustainability in future crises. The clinical implications highlight the need for specific interventions to support the emotional and professional well-being of perioperative nurses, ensuring high quality care and continuity of health services.

## 1. Introduction

Nursing staff shortages and high turnover rates are pressing global challenges that undermine healthcare systems, particularly in Europe and Asia, where they significantly impact healthcare performance [1]. Persistent shortages compromise the delivery of high-quality patient care, leading to adverse outcomes and higher mortality rates [2,3]. These challenges were exacerbated by the COVID-19 pandemic, which imposed an unprecedented strain on healthcare systems, resulting in increased workloads, stress, and burnout rates among nurses.

In Spain, the pandemic exposed long-standing vulnerabilities within the healthcare system, particularly the chronic shortage of nursing staff. Surveys conducted during the pandemic revealed that over 70% of Spanish nurses reported severe burnout, primarily due to unsustainable workloads and insufficient organizational support [4]. These factors, combined with the emotional toll of caring for critically ill patients and high mortality rates, led many nurses to consider leaving their positions. The perioperative nursing sector, in particular, faced unique challenges that are crucial to understand when developing effective strategies to improve retention and workplace conditions.

Perioperative nurses, tasked with performing high-risk procedures while adapting to resource shortages and rapidly evolving protocols, were integral to the pandemic response. Despite their resilience and dedication, these nurses experienced significant psychological distress, including burnout, moral injury, and compassion fatigue [5]. A recent study highlighted the severe emotional and ethical dilemmas faced by perioperative nurses during COVID-19, emphasizing the severe emotional and ethical challenges encountered by these professionals [6]. Prolonged exposure to such conditions led many to resign, further exacerbating workforce shortages in an already strained system.

The workplace in perioperative nursing extends beyond the physical infrastructure to encompass organizational structures, policies, and ethical climates, all of which were significantly altered during the COVID-19 pandemic. Surgical units were restructured into COVID-19 treatment areas, and perioperative nurses were frequently reassigned to intensive care units (ICUs) or placed on reserve for emergency deployment [7,8]. These changes required rapid adaptation, affecting not only clinical responsibilities but also team dynamics, ethical decision making, and overall psychological well-being.

While existing research explores nurses’ ethical challenges and self-care strategies during the pandemic, limited evidence specifically addresses perioperative nurses’ experiences in reorganized workplaces. Prolonged exposure to moral dilemmas, exacerbated by changing roles and high-stress environments, contributed to compassion fatigue, ethical distress, and attrition [6]. These findings highlight the need for resilience-building strategies, leadership support, and ethics training to enhance professional well-being and strengthen retention in perioperative settings [9].

Research has identified multiple factors contributing to nurses’ intentions to leave, including inadequate personal protective equipment (PPE), extended work hours, disrupted social relationships, high stress, and lack of managerial support [10,11]. While the experiences of ICU nurses have been extensively studied [12,13], there is a notable lack of research addressing the specific challenges and decision-making processes of perioperative nurses during the pandemic [14]. This gap in knowledge is critical, as it limits the ability to design targeted interventions to improve workplace conditions and support nurses in this specialized field.

This study seeks to investigate the factors influencing perioperative nurses’ decisions to resign or change roles during or after the COVID-19 pandemic, providing insights into systemic, ethical, and emotional contributors to professional attrition. Unlike quantitative studies that primarily capture numerical trends, this qualitative approach allows nurses to express their experiences in their own words, offering a deeper understanding of their motivations and struggles. By capturing firsthand narratives, this research not only reveals the underlying drivers of workforce instability but also informs the design of more effective, experience-driven interventions. These insights are essential for developing policies and support mechanisms that genuinely resonate with the lived realities of perioperative nurses. In doing so, this study contributes to a broader effort to stabilize the perioperative nursing workforce, improve retention, and enhance resilience in healthcare systems facing ongoing and future crises.

## 2. Materials and Methods

### 2.1. Study Design

This qualitative study employed a phenomenological hermeneutical approach to explore the lived experiences of perioperative nurses who resigned or changed workplaces following the COVID-19 pandemic. The research followed Lindseth and Norberg’s phenomenological hermeneutical approach [15,16], grounded in Ricoeur’s theory of interpretation, which emphasizes the understanding of lived experiences through iterative narrative interpretation [17].

### 2.2. Setting and Participants

This study was conducted at a tertiary-level, high-tech university hospital in central Barcelona, Spain. With over 700 inpatient beds, more than 24 operating rooms, and several intensive care units (ICUs), restructured perioperative services and COVID-19 treatment areas caused significant changes in their professional roles. In this context, nurses, especially perioperative nurses, were reassigned to frontline roles, facing high demand and extreme working conditions, including a lack of resources, the constant risk of infection, and the emotional pressure of treating critically ill patients.

A purposive sampling strategy was employed to recruit perioperative nurses who had resigned, changed workplaces, or taken extended leave (>1 year) following their experiences during the COVID-19 pandemic. The target population consisted of perioperative nurses from the hospital’s surgical units, all of whom had worked during the pandemic. Of 24 invited nurses, six declined (four for personal/scheduling reasons, two due to emotional distress), resulting in 18 participants meeting the following inclusion criteria: (1) working as a perioperative nurse during the COVID-19 pandemic, (2) experiencing a professional transition (resignation, workplace change, or extended leave before returning), and (3) willingness to share experiences. The exclusion criteria included nurses who had not worked in a perioperative setting during the pandemic or were unwilling to participate.

### 2.3. Data Collection

Data were collected between December 2021 and March 2022 through in-depth, semi-structured interviews conducted via videoconferencing to ensure accessibility and participant convenience. This method allowed participants to engage in a familiar environment, facilitating greater openness and honesty in their responses. An interview guide was developed to address key topics, including experiences of working during the pandemic, factors influencing decisions to resign or change workplaces, emotional and ethical challenges encountered, and reflections on organizational support. The interview guide underwent pilot testing to ensure clarity and relevance. See Table 1. The researcher contacted participants directly, explaining the study objectives and inviting them to participate. No gatekeeper was used. Four of those approached declined participation due to personal reasons and scheduling conflicts. Informed consent was obtained before each interview, ensuring ethical compliance. Interviews, which lasted between 45 and 90 min, were audio-recorded with participant consent and subsequently transcribed verbatim.

To foster a trusting relationship with participants and ensure a better understanding of their responses, the interviews were conducted by the author, a researcher with extensive experience in qualitative research (PhD) who is also a nurse with a doctoral degree and previously worked at the same hospital. This pre-existing professional relationship facilitated rapport, but participants were encouraged to share their views independently. To enhance credibility, participants were given the opportunity to review and validate their transcripts for accuracy before analysis commenced. Out of all the potential informants approached, four declined to participate due to personal reasons and scheduling conflicts. This comprehensive and careful approach to data collection not only ensured the quality and reliability of the information obtained but also respected and prioritized the well-being of the participants, ensuring their full engagement and informed consent.

The interviewer had extensive experience in qualitative research, enhancing the credibility and depth of the interviews. Participants were encouraged to express their views independently, and the researcher’s interests in the topic were transparently communicated. This approach allowed for relevant questions and deeper exploration of participants’ experiences. Once thematic redundancy was reached, transcripts were returned to participants for comment and/or correction to ensure validity of the data.

### 2.4. Data Analysis

Data were analyzed using Atlas.ti, version 22 (ATLAS.ti Scientific Software Development GmbH, Berlin, Germany) to systematically manage and code the transcripts.

The analysis followed the phenomenological hermeneutical method described by Lindseth and Norberg (2004), which involved three iterative steps [16]: (1) during the naïve reading phase, transcripts were read multiple times to gain a holistic understanding of the narratives (authors A.S.S. and R.A.M.); (2) structural analysis involved dividing the data into meaning units, which were condensed, coded, and organized into themes and subthemes that captured the participants’ experiences (conducted by M.G.P., S.A.M. and E.M.S.); and (3) comprehensive interpretation to refine the themes reflecting the participants’ lived experiences. Selected excerpts illustrate the participants’ perspectives.

### 2.5. Rigor and Reflexivity

To ensure the study’s rigor, we adhered to Lincoln and Guba’s criteria [18]. Credibility was ensured through prolonged engagement with the data, peer debriefing, and triangulation of data sources and research. Additionally, member checking was employed to validate the interpretations with participants.

Transferability was addressed by providing detailed descriptions of the study context and participants to enable readers to assess applicability to other settings. Dependability and confirmability were established by maintaining an audit trail documenting all analytical processes, ensuring transparency and consistency.

Furthermore, reflexivity was integral to the research process. The researchers engaged in continuous self-examination to acknowledge and mitigate potential biases. A reflexive journal was maintained to document decisions, reflections, and methodological adjustments.

To ensure comprehensive and transparent reporting, we adhered to the Consolidated Criteria for Reporting Qualitative Research (COREQ) guidelines [19]. The COREQ checklist was used as a reference for reliability and is provided as the Appendix A.

### 2.6. Ethical Considerations

The institutional review board approved the study of the university hospital under ethical code IIBSP-COV-2020-58. All participants provided written informed consent before participation. Confidentiality and anonymity were assured throughout the study, with all identifying information removed from transcripts and findings. The study also adhered to the principles outlined in the Declaration of Helsinki (2013) [20].

Ethical measures ensured consent from participants, confidentiality with fictitious names, and secure data handling in compliance with GDPR (Regulation (EU) 2016/679) and Organic Law 3/2018 on data protection. Interviews were recorded, transcribed, reviewed by participants, and subsequently deleted to maintain confidentiality and data security [21].

## 3. Results

All participants in this study were women, with a mean of age of 42.2 years (SD = 9.80), in perioperative nursing. Their work histories reflected diverse career trajectories influenced by the COVID-19 pandemic. Of the 18 participants, nine left the profession entirely, five changed their workplace, and four took a leave of absence before returning after more than a year.

Most participants (70%) worked day shifts, while smaller proportions worked afternoon (15%) or night shifts (10%), and one participant (5%) had an irregular schedule. Most participants (80%) had permanent contracts, whereas 20% were on temporary contracts.

Household composition varied among participants: 45% lived with children, 25% lived alone, 10% lived with an elderly family member, and 20% reported other living arrangements. The pandemic significantly disrupted their professional routines, with 70% being sent to the reserve workforce, 50% reassigned to other units, and 95% directly treating COVID-19 patients. Additionally, 40% of the participants took sick leave due to COVID-19-related health issues. (See Table 2 and Table 3).

Qualitative data analysis revealed four major themes that encapsulating the experiences of perioperative nurses during the COVID-19 pandemic: (1) balancing professional duty and personal limits, (2) the role of workplace culture, (3) resilience and moral conflict, and (4) the emotional cost of caring. These themes shed light on the multifaceted factors that drove many nurses to leave or change roles, as summarized in Table 4. Below, we present the key findings and participant narratives without extensive interpretation. The subsequent discussion offers deeper analysis and situates these results within the existing literature.

### 3.1. Theme 1: Balancing Professional Duty and Personal Limits

This theme captures the internal struggle nurses experienced as they tried to meet the demands of their roles while grappling with their own physical and emotional limitations. Two subthemes emerged:Subtheme 1.1: Initial Motivation to Persevere

Participants expressed a strong sense of duty to their patients and colleagues, which initially motivated them to endure the crisis. Many felt a deep ethical obligation to stay, even when it meant personal sacrifices:

“At the start, I felt like I couldn’t walk away. Patients needed us, and if we didn’t show up, who would? That sense of responsibility kept me going, even when I was exhausted”.(Participant 2)

For some, professional identity and pride underpinned their motivation:

“Nursing is who I am. It’s not just a job; it’s a calling. I couldn’t imagine abandoning my patients, no matter how hard things got”.(Participant 4)

One participant recalled witnessing a colleague collapse from exhaustion, underscoring the severity of the crisis:

“I remember one night when a colleague collapsed in the hallway. We had to keep working because there was no one to replace us. It was terrifying, but we pushed through”.(Participant 5)

Subtheme 1.2: The Breaking Point

As the pandemic wore on, the relentless demands of the surgical area pushed many participants to their limits. The workload, combined with emotional strain, left nurses feeling they could no longer continue:

“I gave everything I had, but it never felt like enough. The workload just kept piling up, and I couldn’t see an end in sight. Eventually, I realized I couldn’t keep going”.(Participant 6)

For some, guilt over not meeting their own care standards was decisive:

“It wasn’t just physical fatigue. It was the guilt of knowing I couldn’t give my patients the care they deserved. That broke me more than anything else”.(Participant 7)

Another nurse shared a pivotal moment when she realized she was emotionally drained and could no longer continue:

“A patient I had been taking care of for days passed away alone, and I was the only person with them. I held their hand and thought, ‘I can’t do this anymore.’ That’s when I decided to leave”.(Participant 9)

### 3.2. Theme 2: The Role of Workplace Culture

Workplace culture significantly influenced participants’ experiences during the pandemic, both positively and negatively. Two subthemes emerged:Subtheme 2.1: Peer Support as a Lifeline

Collegiality was a key source of resilience for many nurses. The support and solidarity of their peers provided emotional and practical relief in an otherwise overwhelming situation:

“The only thing that kept me going was my team. We supported each other in ways management never did. Without them, I wouldn’t have made it”.(Participant 10)

This collegiality acted as a buffer against overwhelming stress:

“We were like a family, stepping in for each other whenever someone needed a break or couldn’t handle the pressure. That bond was the only positive thing during this time”.(Participant 12)

Small gestures, like leaving encouraging notes, also provided a sense of emotional relief:

“We started leaving little notes of encouragement for each other. Just small things like ‘You’re doing great’ or ‘Hang in there’. It sounds silly, but those notes made all the difference”.(Participant 11)

Subtheme 2.2: Leadership’s Role in Burnout

By contrast, participants frequently highlighted how leadership frequently failed to recognize or address frontline challenges:

“Management didn’t understand what we were going through. Decisions were made without consulting us, and we were left to deal with everything on our own”.(Participant 13)

This perceived neglect eroded trust and morale:

“It felt like we were disposable. There was no acknowledgment of what we were going through, and that made it impossible to keep going”.(Participant 6)

One participant described the final straw:

“When I finally spoke up about how exhausted we were, my manager just said, ‘That’s the job.’ I realized then that nothing was going to change”.(Participant 14)

### 3.3. Theme 3: Resilience and Moral Conflict

The pandemic presented nurses with profound ethical dilemmas that tested their professional values and emotional resilience. Two subthemes emerged:Subtheme 3.1: Ethical Dilemmas and Guilt

Participants described the daily moral conflicts they faced, often feeling torn between their duty to patients and their own well-being:

“Every day, I had to decide whether to prioritize my health or my patients. That’s not a choice any nurse should have to make”.(Participant 7)

Some felt guilt at not living up to their standards:

“I wanted to give my best, but I was so drained that I felt like I was doing a disservice to my patients. That guilt ate away at me”.(Participant 16)

One nurse detailed a particularly distressing experience:

“I had to decide who got the last ventilator. It’s a decision I never thought I’d have to make. I still wake up at night thinking about it”.(Participant 18)

Subtheme 3.2: Erosion of Professional Identity

Ongoing moral distress, compounded by inadequate managerial support and limited resources, caused participants to feel disconnected from their roles:

“Nursing used to be my passion, but after everything we went through, I felt like I didn’t recognize myself anymore. It wasn’t what I signed up for”.(Participant 17)

This loss of purpose and fulfilment ultimately drove some to leave:

“I loved being a nurse, but the pandemic stripped that away. I didn’t want to keep doing something that was breaking me”.(Participant 18)

### 3.4. Theme 4: The Emotional Cost of Caring

Working in the surgical area during the pandemic exacted a heavy emotional toll, driven by repetitive exposure to grief and trauma: Two subthemes emerged:Subtheme 4.1: Grief and Emotional Trauma

Participants recounted the profound emotional impact of witnessing patient suffering and family grief. These moments left lasting emotional scars:

“The hardest part was watching families say goodbye over a screen. It felt so wrong, and I couldn’t stop replaying those moments in my mind”.(Participant 11)

Another participant described how supporting patients who had no family took an immense emotional toll:

“I held his hand while he passed away because no one else was there. It felt like an unbearable weight”.(Participant 15)

Continuous exposure to loss often led to emotional trauma:

“Every loss felt personal. I carried it home with me, and it became impossible to separate my work from my life”.(Participant 8)

Subtheme 4.2: Lack of Mental Health Support

Many participants pointed to the absence of mental health resources as a critical factor in their emotional exhaustion. They felt unsupported and isolated in dealing with the psychological burden of their work:

“We were expected to keep going without any real support. No counseling, no debriefing—just go home, sleep, and do it all over again”.(Participant 1)

One nurse described how the lack of institutional support led her to leave the profession:

“I asked for help, and the response was ‘we all have to deal with it.’ But how do you deal with so much death, so much suffering, and just act like nothing happened?”.(Participant 17)

Feeling abandoned and isolated, many concluded that leaving was their only viable way to preserve their own well-being:

“If there had been someone to help us process everything we were going through, maybe I wouldn’t have felt so alone. But there was nothing”.(Participant 2)

## 4. Discussion

This study explored the factors influencing perioperative nurses’ decisions to leave the profession or change workplaces following the COVID-19 pandemic. The findings revealed complex interactions between systemic, ethical, and emotional factors that shaped these decisions. These results provide valuable insights into the lived experiences of nurses during an unprecedented global health crisis and highlight critical areas for intervention to improve workforce retention.

Balancing Professional Duty and Personal Limits. The tension between professional duty and personal limits was a central theme, with participants describing feelings of moral obligation despite experiencing physical and emotional exhaustion. This aligns with previous studies that highlighted how moral distress arises when nurses are unable to provide the quality of care they aspire to due to systemic constraints [22]. Similarly, it has been found that ICU nurses often experience burnout linked to moral distress, which may lead to professional attrition [23].

However, our findings expand on this by emphasizing that the prolonged demands of the pandemic pushed nurses beyond their limits, making resignation or workplace transitions a necessary step for self-preservation. While although some authors advocate for resilience-building programs to mitigate burnout, our study suggests that systemic issues, such as inadequate staffing and overwhelming workloads, must also be addressed to prevent such outcomes [24].

The Role of Workplace Culture. Workplace culture was pivotal in shaping nurses’ experiences. Peer support provided emotional relief and fostered resilience, which is consistent with previous findings that underscored the protective effect of collegial relationships during crises [25]. Participants in our study also described a lack of empathetic leadership, which exacerbated their feelings of disconnection and undervaluation. It is argued that transformational leadership, characterized by empathy and inclusiveness, can reduce burnout and increase retention, yet our findings reveal that many nurses experienced leadership failures that undermined their capacity to cope [26].

This neglect of frontline workers was also pointed out in previous studies that emphasized the importance of empowering workplace environments in fostering job satisfaction [27]. Our study corroborates this perspective, suggesting that poor managerial engagement may have been a decisive factor in participants’ decisions to leave their roles.

Resilience and Moral Conflict. Ethical dilemmas emerged as a significant factor, with nurses facing decisions that placed their personal safety at odds with their professional values. A previous study described similar challenges during the pandemic, where healthcare workers had to balance their ethical obligations with unprecedented risks [28]. Participants in this study expressed feelings of guilt and disillusionment, highlighting the emotional toll of these dilemmas.

It is argued that ethical frameworks can provide healthcare workers with guidance during crises, reducing the emotional burden of moral conflict [29]. However, our findings suggest that such frameworks were either absent or insufficiently implemented, leaving nurses to navigate these dilemmas in isolation. This underscores the need for clearer ethical support systems in healthcare settings.

The Emotional Cost of Caring. Prolonged exposure to suffering and death had a profound emotional impact on participants, consistent with previous studies that documented the psychological toll of secondary traumatic stress among healthcare workers [30]. Many participants described feelings of helplessness and emotional depletion, compounded by a lack of mental health resources. The importance of accessible mental health interventions has also been highlighted to address the psychological impact of the pandemic, which our findings support [31].

Participants’ experiences suggest that the absence of formal mental health support left them isolated in processing their trauma. Institutionalizing counselling services and regular debriefing sessions could mitigate these effects and provide a pathway for recovery.

### 4.1. Limitations and Strenghts

While this study provides valuable insights, several limitations should be acknowledged. The research was conducted in a single tertiary-level hospital in Spain, which may limit the generalizability of findings to other healthcare systems. Given the qualitative nature of the study, findings rely on self-reported experiences, which could introduce recall bias and individual differences in perception. Additionally, participant recruitment was based on voluntary participation, which may have led to an overrepresentation of nurses who experienced particularly strong ethical and emotional conflicts. Another limitation is the study’s focus on workplace restructuring and ethical challenges, without an in-depth analysis of hospital architecture, medical technology, or spatial constraints, which may have also influenced nurses’ fatigue and professional difficulties. Despite these limitations, the study had several strengths. The inclusion of multiple perspectives and the triangulation of data through detailed interviews strengthened the validity of the results. The diversity of experiences captured in the study provided a comprehensive view of the challenges faced by the nurses, which could inform future interventions and policies to improve the retention and well-being of healthcare staff. Additionally, using the hermeneutic phenomenological method allowed for a deep and nuanced understanding of perioperative nurses’ experiences during the COVID-19 pandemic, providing a rich narrative that captured the complexity of the factors influencing their professional decisions. Finally, the study addressed a critical gap in the literature by exploring the specific experiences of perioperative nurses. This group had received less attention in previous research on the impact of the pandemic on healthcare staff, adding unique and relevant value to the findings.

### 4.2. Implications for Practice

The findings highlight critical areas for systemic and cultural reform within healthcare organizations to improve workforce retention. Leadership development should prioritize transformacional leadership training that fosters empathy, inclusiveness, and transparent communication to foster trust and resilience among nurses. Enhancing mental health resources is essential, including institutionalizing accessible mental health services, structured debriefing sessions and peer support programs to help nurses cope with the emotional and ethical challenges of their work. Addressing workload issues by ensuring adequate staffing and equitable workload distribution are crucial to reducing burnout and preventing moral distress. Additionally, strengthening ethical support mechanisms through clear ethical frameworks and open discussion forums will help nurses to navigate moral conflicts with confidence, reducing professional disillusionment and attrition.

## 5. Conclusions

This study highlights the systemic, ethical, and emotional challenges faced by perioperative nurses during the COVID-19 pandemic and their influence on decisions to leave the profession or change workplaces. Addressing these challenges through ethical leadership, mental health support, and structured workforce planning to enhance nurse retention and well-being o ensure long-term improvements, leadership training should be strengthened, workplace mental health initiatives institutionalized, and ethical decision-making frameworks expanded to support nurses facing moral dilemmas.

The COVID-19 pandemic exposed critical vulnerabilities in perioperative nursing, emphasizing the urgent need for systemic reforms to improve workplace stability, ethical support, and mental health resources. Addressing these challenges is essential for retaining nurses, ensuring workforce resilience, and maintaining the stability of healthcare systems in future health crises. Healthcare organizations must prioritize the well-being and professional sustainability of perioperative nurses to maintain workforce stability, ensure high-quality patient care, and prepare for future healthcare emergencies.

Future research should explore the long-term psychological impact of workplace reassignments on perioperative nurses, particularly in relation to burnout, moral distress, and career trajectories. Additionally, investigating the role of the built environment—including spatial constraints, ICU conversions, and resource availability—could provide insight into how physical infrastructure affects nurse well-being and decision-making stress. Comparative studies across different healthcare systems and institutional policies are also needed to identify best practices for workforce retention, ethical support mechanisms, and crisis preparedness strategies that can be implemented in future public health emergencies.

## Figures and Tables

**Table 1 healthcare-13-00391-t001:** Specific interview questions.

“Can you describe how your experiences in the perioperative area or surgical area during the pandemic influenced your career decisions?”
“What were the key factors that shaped your working conditions and challenges during this time?”
“How did your feelings and emotions evolve as you contemplated leaving your role in the perioperative or surgical area?”
“Did you encounter external pressures to remain in the perioperative or surgical area? If yes, who applied these pressures and in what manner?”
“Reflecting on your experience, what specific measures or actions might have convinced you to stay in the perioperative or surgical area?”
“What advice would you offer to healthcare workers in perioperative or surgical roles who may face similar challenges during future pandemics?”

**Table 2 healthcare-13-00391-t002:** Characteristics of the sample.

Characteristic	Subcategory	Mean (SD)	Frequency (%)
Gender	Female		100% (18)
Age		42.2 (9.80)	
Household composition			
Living alone	Living alone		25% (5)
Living with children	Living with children		45% (8)
Living with elder	Living with elder		10% (2)
Other	Other		20% (3)
Years of experience		18.5 (8.90)	
Working shift			
Day shift	Day shift		70% (13)
Afternoon shift	Afternoon shift		15% (3)
Night shift	Night shift		10% (2)
Other	Other		5% (1)
Type of contract			
Permanent	Permanent		80% (14)
Temporary	Temporary		20% (4)
Type of work during the pandemic			
	Sent to the reserve		70% (13)
Reassigned to other units	Reassigned to other units		50% (9)
Treated COVID-19 patients	Treated COVID-19 patients		95% (17)
COVID-19-related sick leave	COVID-19-related sick leave		40% (7)

All data are expressed as the mean (SD) or frequency (percentage), as appropriate. %: percentage; SD: standard deviation.

**Table 3 healthcare-13-00391-t003:** Participant profile.

Participant	Age	Household Composition	Work Shift	Contract Type	Work Reassignment During Pandemic	Decision After Pandemic
P1	40	Lives alone	Rotating shift	Permanent	Reassigned to ICU	Changed workplace
P2	35	Lives with children	Day shift	Temporary	Sent to reserve	Left profession
P3	45	Lives with children	Day shift	Permanent	Treated COVID-19 patients	Took extended leave
P4	50	Lives with children	Rotating shift	Permanent	Reassigned to ICU	Returned after leave
P5	38	Lives alone	Night shift	Temporary	Sent to reserve	Left profession
P6	42	Lives with elderly parent	Afternoon shift	Permanent	Treated COVID-19 patients	Changed workplace
P7	37	Lives with children	Day shift	Permanent	Reassigned to ICU	Left profession
P8	41	Lives with elderly parent	Rotating shift	Permanent	Treated COVID-19 patients	Changed workplace
P9	39	Lives alone	Day shift	Temporary	Sent to reserve	Left profession
P10	44	Lives with children	Day shift	Permanent	Treated COVID-19 patients	Returned after leave
P11	36	Lives alone	Rotating shift	Permanent	Sent to reserve	Changed workplace
P12	48	Lives with children	Day shift	Permanent	Reassigned to ICU	Left profession
P13	43	Lives with children	Afternoon shift	Temporary	Treated COVID-19 patients	Returned after leave
P14	47	Lives alone	Day shift	Permanent	Sent to reserve	Changed workplace
P15	39	Lives alone	Night shift	Temporary	Reassigned to ICU	Left profession
P16	46	Lives with elderly parent	Rotating shift	Permanent	Treated COVID-19 patients	Returned after leave
P17	38	Lives with children	Day shift	Permanent	Reassigned to ICU	Changed workplace
P18	49	Lives with children	Day shift	Temporary	Treated COVID-19 patients	Left profession

**Table 4 healthcare-13-00391-t004:** Interconnected Themes and Subthemes.

Themes	Subthemes	Key Points
1. Balancing Professional Duty and Personal Limits	1.1. Initial Motivation to Persevere 1.2. The Breaking Point	Struggles with physical and emotional limits; tension between duty and self care.
2. The Role of Workplace Culture	2.1. Peer Support as a Lifeline 2.2. Leadership’s Role in Burnout	Impact of peer support versus lack of managerial empathy; trust erosion.
3. Resilience and Moral Conflict	3.1. Ethical Dilemmas and Guilt 3.2. Erosion of Professional Identity	Moral dilemmas in prioritizing patient care vs. personal safety; identity erosion.
4. The Emotional Cost of Caring	4.1. Grief and Emotional Trauma 4.2. Lack of Mental Health Support	Emotional toll from trauma and grief; inadequate mental health resources.

## Data Availability

The datasets generated and/or analyzed during the current study are not publicly available but are available from the corresponding author upon reasonable requests.

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
