# Peer review of "Shifting Compasses: A Qualitative Study of Lived Experiences Driving Perioperative Nurses to Leave the Profession Post COVID-19"

_healthcare, 2025, doi:10.3390/healthcare13040391_

Round 1

Reviewer 1 Report

Comments and Suggestions for Authors

The study's subject is very relevant to health organization activities. In addition, it is a very topical subject, even though time has passed on the COVID-19 pandemic. Nurses are a key professional group in medical crises, and research into their safety and well-being at work should be conducted across a wide range of topics. Notably, the original statements of the participants in the survey capture emotions far beyond quantitative research.

The Abstract is worded correctly, although parts of the Conclusion should be reworded if the authors include further comments in the review.

The Introduction section lacks a definition of Workplace, which the researchers address. Analysis of the article indicates that the research did not attempt to identify the characteristics of the built environment (architecture, medical technology) in which the nurses worked and the impact of spatial strictures during the pandemic on their fatigue and difficulties in daily activities.

The background of the study was the work organization in the hospital—if so, it should be described in more detail. The additional duties that affected perioperative nurses should also be mentioned.

The Materials and Methods section should include information about the hospital where the nurses involved in the study worked, including its location, size, and role during the pandemic.

The Characteristics of the Sample table should be separated into two parts: demographic characteristics and type of work during the pandemic. The data is given illegibly, e.g., Gender - 100%.

The content of the Conclusion should be significantly expanded. What are the application guidelines? What research still needs to be done? There should be a clearer reference to the study's limitations and future research directions proposed by the study authors.

Reviewer 2 Report

Comments and Suggestions for Authors

This is a capably conducted study on an important health-related topic of nursing retention post-COVID. I have the following suggestions for improvement.

1. The Introduction could be enhanced with a bit more attention to the significance of the study. The single sentence, "By examining these experiences..." is a bit terse. In a few additional sentences, expound on the merits of this specific study. For example, collecting data from the nurses featured here in their own words may provide avenues for more effective interventions that resonate with their experiences, as compared with more quantitative approaches. 

2. Please say a bit about the hospital in which the nurses worked. That context seems important. Also, why was this particular hospital chosen?

3. Please identify A.S.S. as an author on line 121. Just say "author" without providing the name. Otherwise, the abbreviation is unclear.

4. The Results section has some great quotes, but I'd welcome another one or two for themes and sub-themes where the authors have them available. Doing so would more clearly reveal patterns. Also, if there are a few examples of more extended narratives (experiences recounted as stories), those would be interesting. i do think this section could be built out a bit more. 

5. Please alert the reader that extensive researcher interpretations are reserved for the Discussion section. Some Results sections have a great deal of interpretation in them. This paper does not, which is fine. But please make that clear to set the reader's expectations.

6. Some recommendations for future research would be an excellent addition. 

Well done, with some room for improvement.

Reviewer 3 Report

Comments and Suggestions for Authors

Dear authors, the revised manuscript addresses a Qualitative Study of Lived Experiences Driving Perioperative Nurses to Leave the Profession Post- COVID-19. It is an interesting approach to a specific group of nurses in Spain. The comments I will make are based on the COREQ guidelines for reporting qualitative studies.

Title: is suitable for a qualitative study.

Abstract: is structured and informative. In the methodology section, more information on the method should be provided (add information on the analysis process, software, ethical aspects) and not refer to the results (the number of participating nurses should be in the results). The conclusions should be more precise in order to provide answers to the objective of the research taking into account the results achieved. In addition to addressing clinical implications of the research.

Keywords: The number is excessive. They should be limited to a maximum of 6. The concepts used should conform as far as possible to DeCS/MeSH terms, e.g. for Qualitative Study the correct descriptor is "Qualitative Research" (do not duplicate with the term phenomenology).

Introduction: The introduction is adequate and well structured, ending with the objective of the research.

Material and methods:

Study design: revise the citation to the authors Lindseth and Norberg (2004, 2022) and remove the dates so as not to confuse APA style.

A ‘setting’ sub-section should be included to contextualise the social and labour context of Spain in relation to the figure of nurses and, specifically, of the hospital where the study was carried out.

Participants: in this section it is more correct to give an overview of the target population likely to become participants (number of nurses, characteristics of their profiles...). It is possible to indicate if a greater number of nurses were contacted than those who finally made up the sample, but it is not correct to refer to the socio-demographic characteristics of the participants (this information should be included in the results). In the event that nurses were contacted who did not participate in the study, it is a COREQ criterion of rigour to comment on the reasons. Any participant refused to participate or dropped out? Explain more clearly the inclusion and exclusion criteria. Better specify the type of sampling (convenience, theoretical, both...).

Table 1 should be translated into results. In any case, it is recommended to modify the table to implement another one more suitable for qualitative studies. It is not so relevant the gender percentage, the average age of 18 people, the percentage of nurses without children, or the percentage of nurses on night shift... In a qualitative study it is more relevant to know that participant 1 is a woman, who is 40 years old, lives alone, works a rotating shift and has a permanent contract. What is of interest with this table is to be able to link appointments to a specific participant profile. Revise this table according to the recommendations to give more value to the socio-demographic results.

Data collection: The datand process of analysis is mix in, which is described in the following sub-sections in duplicated form. I refer to credibility: transcripts returned to informants for verification. How participants were contacted needs to be specified (was a gatekeeper used, which researcher, was there prior researcher contact with participants?. information on the analysis process (identification of verbatims in the interviews after transcription) is also mixed in.

Data analysis: Explain more precisely the coding process; was an intentional or emergent category analysis carried out? What software was used for the transcriptions? Do not duplicate information about the software: "...software Atlas.ti software..."

Rigor and Trustworthiness: In qualitative research it is more appropriate to use concepts such as rigor and reflexivity for the heading, rather than trustworthiness. Within this section, the criteria of trustworthiness (which in qualitative research is more correctly referred to as credibility) should be addressed. Was there any triangulation of data, of researchers?

It is not necessary to explain what the COREQ results reporting guideline consists of: "The COREQ checklist, consisting of 32 items for interviews and focus groups, was used as a reference for reliability and is provided as supplementary material"

Results:

Start the results with a description of the characteristics of the participants.

Table 3 with the themes and sub-themes is best placed before starting the content analysis of each of the themes so that the reader can better situate himself/herself.

Discussion: Check citation style (do not use APA).

Conclussions: As I commented in the abstract, the conclusions should be more precise in order to provide answers to the objective of the research taking into account the results achieved. In addition to addressing clinical implications of the research.

References: Review the MDPI style in the references.

Round 2

Reviewer 2 Report

Comments and Suggestions for Authors

I commend the authors on an excellent revision. I have no additional revisions to recommend. Well done!